# Evaluation of the Azure Kinect and Its Comparison to Kinect V1 and Kinect V2

**DOI:** 10.3390/s21020413

**Published:** 2021-01-08

**Authors:** Michal Tölgyessy, Martin Dekan, Ľuboš Chovanec, Peter Hubinský

**Affiliations:** Institute of Robotics and Cybernetics, Faculty of Electrical Engineering and Information Technology STU in Bratislava, Ilkovičova 3, 812 19 Bratislava, Slovakia; martin.dekan@stuba.sk (M.D.); lubos.chovanec@stuba.sk (Ľ.C.); peter.hubinsky@stuba.sk (P.H.)

**Keywords:** Kinect, Azure Kinect, robotics, mapping, SLAM (simultaneous localization and mapping), HRI (human–robot interaction), 3D scanning, depth imaging, object recognition, gesture recognition

## Abstract

The Azure Kinect is the successor of Kinect v1 and Kinect v2. In this paper we perform brief data analysis and comparison of all Kinect versions with focus on precision (repeatability) and various aspects of noise of these three sensors. Then we thoroughly evaluate the new Azure Kinect; namely its warm-up time, precision (and sources of its variability), accuracy (thoroughly, using a robotic arm), reflectivity (using 18 different materials), and the multipath and flying pixel phenomenon. Furthermore, we validate its performance in both indoor and outdoor environments, including direct and indirect sun conditions. We conclude with a discussion on its improvements in the context of the evolution of the Kinect sensor. It was shown that it is crucial to choose well designed experiments to measure accuracy, since the RGB and depth camera are not aligned. Our measurements confirm the officially stated values, namely standard deviation ≤17 mm, and distance error <11 mm in up to 3.5 m distance from the sensor in all four supported modes. The device, however, has to be warmed up for at least 40–50 min to give stable results. Due to the time-of-flight technology, the Azure Kinect cannot be reliably used in direct sunlight. Therefore, it is convenient mostly for indoor applications.

## 1. Introduction

The Kinect Xbox 360 has been a revolution in affordable 3D sensing. Initially meant only for the gaming industry, it was soon to be used by scientists, robotics enthusiasts and hobbyists all around the world. It was later followed by the release of another Kinect—Kinect for Windows. We will refer to the former as Kinect v1, and to the latter as Kinect v2. Both versions have been widely used by the research community in various scientific such as object detection and object recognition [1,2,3], mapping and SLAM [4,5,6], gesture recognition and human–machine interaction (HMI) [7,8,9], telepresence [10,11], virtual reality, mixed reality, and medicine and rehabilitation [12,13,14,15,16]. According to [17] there have been hundreds of papers written and published on this subject. However, both sensors are now discontinued and are no longer being officially distributed and sold. In 2019 Microsoft released the Azure Kinect, which is no longer meant for the gaming market in any way; it is promoted as a developer kit with advanced AI sensors for building computer vision and speech models. Therefore, we focus on the analysis and evaluation of this sensor and the depth image data it produces.

Our paper is organized as follows. Firstly, we describe the relevant features of each of the three sensors—Kinect v1, Kinect v2 and Azure Kinect (Figure 1). Then we briefly compare the output of all Kinects. Our focus is not set on complex evaluation of previous Kinect versions as this has been done before (for reference see [18,19,20,21,22,23,24]). In the last section, we focus primarily on the Azure Kinect and thoroughly evaluate its performance, namely:Warm-up time (the effect of device temperature on its precision)AccuracyPrecisionColor and material effect on sensor performancePrecision variability analysisPerformance in outdoor environment

## 2. Kinects’ Specifications

Both earlier versions of the Kinect have one depth camera and one color camera. The Kinect v1 measures depth with the pattern projection principle, where a known infrared pattern is projected onto the scene and out of its distortion the depth is computed. The Kinect v2 utilizes the continuous wave (CW) intensity modulation approach, which is most commonly used in time-of-flight (ToF) cameras [18].

In a continuous-wave (CW) time-of-flight (ToF) camera, light from an amplitude modulated light source is backscattered by objects in the camera’s field of view, and the phase delay of the amplitude envelope is measured between the emitted and reflected light. This phase difference is translated into a distance value for each pixel in the imaging array [25].

The Azure Kinect is also based on a CW ToF camera; it uses the image sensor presented in [25]. Unlike Kinect v1 and v2, it supports multiple depth sensing modes and the color camera supports a resolution up to 3840 × 2160 pixels.

The design of the Azure Kinect is shown in Figure 2.

Comparison of the key features of all three Kinects is in Table 1. All data regarding Azure Kinect is taken from the official online documentation.

It works in four different modes-NFOV (narrow field-of-view depth mode) unbinned, WFOV (wide field-of-view depth mode) unbinned, NFOV binned, and WFOV binned. The Azure Kinect has both, a depth camera and an RGB camera; spatial orientation of the RGB image frame and depth image frame is not identical, there is a 1.3-degree difference. The SDK contains convenience functions for the transformation. These two parts are, according to the SDK, time synchronized by the Azure.

## 3. Comparison of all Kinect Versions

In this set of experiments, we focused primarily on the precision of the examined sensors. They were placed on a construction facing a white wall as shown in Figure 3. We measured depth data in three locations (80, 150 and 300 cm), and switched the sensors at the top of the construction so that only one sensor faced the wall during measurement in order to eliminate interferential noise. The three locations were chosen in order to safely capture valid data for approximate min, mid and max range for each of the sensors. A standard measuring tape was used, since in this experiment we measured only precision (repeatability). A total of 1100 frames were captured for every position.

### 3.1. Typical Sensor Data

Typical depth data of tested sensors is presented in Figure 4.

As can be seen, there is a big difference between the first two generations of the Kinect and the new Azure version. The first two cover the whole rectangular area of the pixel matrix with valid data, while the new sensor has hex area for the narrow mode (NFOV), and circular area for the wide mode (WFOV). The data from the Azure Kinect still comes as a matrix, so there are many pixels that are guaranteed to be without information for every measurement. Furthermore, the field of view of the Azure Kinect is wider.

In our experiments we focused on noise-to-distance correlation, and noise relative to object reflectivity.

### 3.2. Experiment No. 1–Noise

In this experiment we focused on the repeatability of the measurement. Typical noise of all three sensors at 800 mm distance can be seen in Figure 5, Figure 6, Figure 7 and Figure 8. The depicted visualization represents standard deviation for each pixel position computed from repeated measurements in the same sensor position (calculated from distance in mm). For better visual clarity, we limited the maximal standard deviation for each sensor/distance to the value that was located in the area of correct measurements. Extreme values of standard deviation we omitted were caused by one of the following-end of sensor range or border between two objects.

As can be seen in presented Figure 5, Figure 6, Figure 7 and Figure 8, the noise of Kinect v2 and Azure Kinect raises at the edges of useful data while the Kinect v1 noise has many areas with high noise. This was expectable as both, Kinect v2 and Azure, work on the same measuring principle.

For the noise to distance correlation, we made measurements at different distances from a white wall. We took an area of 7 × 7 pixels around the center of the captured depth image as this is the area where the measured distance corresponds to the smallest actual Euclidian distance between the sensor chip and the wall; as the wall is perpendicular to the sensor (Figure 9). From these data we calculated the standard deviation for all 49 pixels, and then computed its mean value. From Table 2, it is obvious that Microsoft made progress with their new sensor and the repeatability of the new sensor is much better than the first Kinect generation, and even surpasses the Kinect v2 in 3 out of 4 modes.

### 3.3. Experiment No. 2–Noise

The second experiment focused on the noise-to-reflectivity correspondence. We put a white-board and a cork-board table in the sensing area (Figure 10), and then we checked whether it can be seen in the noise data. All sensors were placed in the same position and distance from the wall, but the presented depth image sizes slightly differ, since each sensor has different resolution.

What we expected was, that for Kinect v1, the board should be nonvisible and for Kinect v2, a change in the data should be detectable. The noise data for Kinect v1 and v2 is in Figure 11 and Figure 12.

As can be seen in Figure 12, there is an outline of the board visible in the Kinect v2 data (for better visual clarity the range of the noise was limited, and only the relevant area of the depth image is shown).

We preformed the same experiment with the Azure Kinect (Figure 13). The board is visible in all 4 modes; therefore, we assume the behavior of Azure in this regard is the same as for Kinect v2. For more information regarding this behavior for Kinect v2 please refer to [26], which deals with this issue in more detail. 

## 4. Evaluation of the Azure Kinect

The Azure Kinect provides multiple settings for frame rate. It can be set to 5, 15 or 30 hz, with the exception of both the unbinned wide depth mode, and RGB 3072px mode, where the 30 fps is not supported. It has a hardware trigger, with fairly stable frame-rate. We recorded 4500 frames with timestamps and found only small fluctuation (±2 ms between frames), which was probably caused by the host PC, not the Azure Kinect itself.

### 4.1. Warm-up Time

The Azure Kinect does not give a stable output value after it is turned on. It needs to be warmed up for some time for the output to stabilize. Therefore, we devised an experiment to determine this time. For this experiment, we put the Azure Kinect roughly 90 cm away from a white wall and started measuring right away with a cold Azure Kinect. We ran the measurement for 80 min and computed the average distance and standard deviation from first 15 s of each minute. From these data we took one center pixel; the results are in Figure 14 and Figure 15. As can be seen, the standard deviation did not change considerably, but the measured distance grew until it stabilized on a value 2 mm higher compared to the starting value. From the results we conclude it is necessary to run the Azure Kinect for at least 60 min to get stabilized output.

Every other experiment was performed with a warmed-up device.

### 4.2. Accuracy

For accuracy measurements, we mounted a white reflective plate to the end effector of a robotic manipulator—ABB IRB4600 (Figure 16 and Figure 17). The goal was to position the plate in precise locations and measure the distance to this plate. The absolute positioning accuracy of the robot end effector is within 0.02 mm according to the ABB IRB4600 datasheet.

The Azure Kinect depth frame is rotated a few degrees from the RGB frame, thus we had to properly align our measuring plate with this frame. Otherwise, the range error would be distorted. First step in this process was to align the plate with the depth frame; for a coarse alignment we used an external IMU sensor (DXL360s with resolution 0.01 deg). Then, for fine tuning, we changed the orientation of the plate, so that 25 selected depth points reported roughly the same average distance. The average for every point did not deviate more than 0.1 mm (Figure 18).

After that, we had to determine the axis along which the robot should move, so that it would not deviate from the depth frame axis (Figure 16). For this purpose, we used both the depth image and the infrared image. At the center of the plane, a bolt head was located. This bolt was clearly visible in depth images at short distances and in infrared images at longer distances. We positioned the robot in such a way, that this bolt was located in the center of the respective image at two different distances. One distance was set to 42 cm and the other to over 2.7 m. These two points defined the line (axis) along which we moved the plate. We assured the bolt was located in the center for multiple measuring positions. Therefore, the axis of robot movement did not deviate from the Z axis of the depth camera for more than 0.09 degrees, which should correspond to sub mm error in reported distance of the Azure Kinect.

Unfortunately, the exact position of the origin of the depth frame is not known. For this reason, we assumed one distance as correct, and all the errors are reported relatively to this position. The originating distance was set to 50 cm, as reported by the unbinned narrow mode of the Azure Kinect (this assumption may be wrong, as it is possible, that the sensor does not report correct distance in any mode or any location). What the reader should take from this experiment is how the accuracy changes with varying distances.

We performed measurements for all 4 modes for distances ranging from 500 mm to 3250 mm with a 250 mm step, and at 3400 mm, which was the last distance our robot could reach. From each measurement, we took 25 points (selected manually similarly to Figure 18) and computed the average distance. The exact position of the points varied for each mode and plate location, but we tried to make them evenly distributed throughout the measurement plate. For the starting 500 mm distance and unbinned narrow mode, the points are the same as shown in Figure 18.

As can be seen in Figure 19, at this range, the distance error does not deviate much, neither with changing modes nor with distances. There is a drop of 8 mm for binned narrow mode, but this is well within datasheet values. If we selected a different distance as the correct one, the error would be even less visible.

### 4.3. Precision

To determine the precision (repeatability) we used the same data as for the accuracy measurements; the resulting precision is shown in Figure 20. As expected, the binned versions of both fields of view show much better results.

### 4.4. Reflectivity

For this test, our aim was to compare the precision of the Azure Kinect for multiple types of material specimens; therefore, they had different reflectivity properties. The specimens and their layouts are shown in Figure 21; the basic reflectivity features can be seen in the infrared image as shown in Figure 22.

We repeated the measurement 300 times and calculated the average and standard deviation. The results are presented in Figure 23 and Figure 24.

As can be seen, there is a correlation between the standard deviation and reflectivity. The less reflective materials have higher standard deviation. An exception was the aluminum thermofoil (bottom right), which had a mirroring effect, thus resulting in higher standard deviation and average distance. What can be seen in the distance data is, that some types of materials affect the reported distances. These materials are either fuzzy, porous or partially transparent.

### 4.5. Precision Variability Analysis

The Azure Kinect works in 4 different modes, what greatly enhances the variability of this sensor. While the narrow mode (NFOV) is great at higher precision scanning of smaller objects for its small noise and angular range, for many applications such as movement detection, mobile robot navigation or mapping, the wide mode (WFOV) provides unrivaled angular range to other RGBD cameras in this price range. From our experiments, we concluded that in the latter mode the data contains some peculiarities which should be reckoned. As can be seen in Figure 8, the noise of the data raises from the center in every direction, making it considerably higher at the end of the range. We suspected that the source of this rise could be due to one or all of these three aspects:The rise of the noise is due to distance. Even though the reported distance is approximately the same, the actual Euclidian distance from the sensor chip is considerably higher (Figure 9).The sensor measures better at the center of the image. This could be due to optical aberration of the lens.The relative angle between the wall and sensor. This angle changes from center to the edges changing the amount of reflected light back to the sensor, which could affect the measurement quality.

The source of the worsened quality could be either of these factors, even all three combined; therefore, we devised an experiment specifically for each of these factors.

To explore the first aspect, we analyzed the standard deviation computed from data acquired by measuring a wall located 1.2 m away from the sensor as shown in Figure 25. The blue curve is the standard deviation of the distance from the plane; the orange curve is the same curve compensated with the real Euclidian distance from the sensor. When comparing the curves, it is clear that the growing Euclidian distance from the lens has direct impact on the noise. Only the extreme left and right areas indicate growing noise.

To test the aberration of the lens, we placed the Azure at one spot 100 cm away from the wall and rotated the sensor about its optical center for every measurement. At each position we made multiple measurements (1100) and calculated the standard deviation. We focused on the same spot on the wall. This way, the distance from the wall at the area of interest did not change; the relative angle between the wall and the area of interest captured by the sensor chip remained constant as well (due to parabolic properties of the lens). The result can be seen in the Table 3.

As can be seen, there is no significant difference between measurements, so we concluded there is no drop in quality of the measurement in respect to the angular position. But there is a considerable drop of quality at the edge of the angular range. But this can be seen on all versions of the Kinect.

The third sub-experiment was aimed at the measurement quality with respect to the relative angle between the wall and the sensor; for that we used the same data as in the first sub-experiment. Our point was to focus on 3D points with the same Euclidian distance from the sensor, but with different angle between the wall and the chip (Figure 9).

From all measured positions, we selected three adjacent depth image rows located close to the center of the image and computed the standard deviation. Then we computed the average of each triplet; this resulted in one dataset (comprising of 8 lines-one for each distance) for further processing. For each point of the final dataset, we stored its distance from the wall, angle between the sensor and the wall and its standard deviation (Figure 26).

Figure 27 and Figure 28 presents the same dataset without extreme values that are irrelevant for further processing. It is clear that there is a correlation between relative angle of measurement and its quality.

The rotation of previous figure denotes the dependence even more (Figure 29).

To highlight these dependences even more, Figure 30 depicts the standard deviation for particular identical distances with respect to the angle for which we measured.

Thus, we concluded that the relative angle between the sensor and the measured object plays an important role in the quality of measurement.

### 4.6. Performance in Outdoor Environment

In the final set of experiments, we examined the Azure Kinect’s performance in outdoor environment. We designed two different scenarios. In the first one, the sensor was facing the sun directly while scanning the test plate (Figure 31). In the second one, the sun was outside sensor’s field of view while shining directly on the test plate (Figure 32). We computed the standard deviation for NFOV and WFOV binned modes on 300 depth images and limited the result to 200 mm for better visibility (Figure 33 and Figure 34 report the results for the first experiment, Figure 35 and Figure 36 report the results for the second experiment).

As can be seen from the figures, in all measurements there is noise in air around the test plate despite the fact there was no substantial dust present. Both WFOV mode measurements are extremely noisy, what makes this mode unusable for outdoor environment. NFOV mode shows much better results; this is most likely caused by the fact that different projectors are used for each of the modes. Surprisingly, the direct sun itself causes no substantial chip flare outside its exact location in the image.

Even though the NFOV mode gave much better results, the range of acceptable data was only within 1.5 m distance. Even though the test plate shows little noise there were many false measurements (approximately 0.3%), which did not happen in indoor environment at all. With growing distance false measurement count grows rapidly. Therefore, we conclude that even the NFOV binned mode usability in outdoor environment is highly limited.

### 4.7. Multipath and Flying Pixel

As stated in the official online documentation, the Azure Kinect suffers from multipath interference. For example, in corners, the IR light from the sensor is reflected off one wall onto the other. This results in invalidated pixels. Similarly, at the edges of objects, pixels can contain mixed signal from foreground and background. This phenomenon is known as the flying pixel problem. To demonstrate this, we put a plate 4 mm thick in front of a wall and focused on the data acquired around the edges of the plate. As can be seen in Figure 37 and Figure 38, the depth data located at the edges of the plate are inaccurately placed outside the actual object.

## 5. Conclusions

We performed series of experiments to thoroughly evaluate the new Azure Kinect. The first set of experiments put the Azure in context with its predecessors, and in the light of our experiments it can be said that in terms of precision (repeatability) its performance is better than both previous versions.

By examining the warm-up time, we came to the conclusion that it shows the same behavior as Kinect v2 and needs to be properly warmed up for about 50–60 min to give stable output. We examined different materials and their reflectivity. We determined the precision and accuracy of the Azure Kinect and discussed why the precision varies.

All our measurements confirm that both, the standard deviation, and systematic error of the Azure Kinect are within the values specified by the official documentation: Standard deviation ≤ 17 mm.Distance error < 11 mm + 0.1% of distance without multi-path interference

The obvious pros and cons of the Azure Kinect are the following:PROS
Half the weight of Kinect v2No need for power supply (lower weight and greater ease of installation)Greater variability–four different modesBetter angular resolutionLower noiseGood accuracyCONS
Object reflectivity issues due to ToF technologyVirtually unusable in outdoor environmentRelatively long warm-up time (at least 40–50 min)Multipath and flying pixel phenomenon

To conclude, the Azure Kinect is a promising small and versatile device with a wide range of uses ranging from object recognition, object reconstruction, mobile robot mapping, navigation and obstacle avoidance, SLAM, to object tracking, people tracking and detection, HCI (human–computer interaction), HMI (human–machine interaction), HRI (human–robot interaction), gesture recognition, virtual reality, telepresence, medical examination, biometry, people detection and identification, and more.

## Figures and Tables

**Figure 1 sensors-21-00413-f001:**
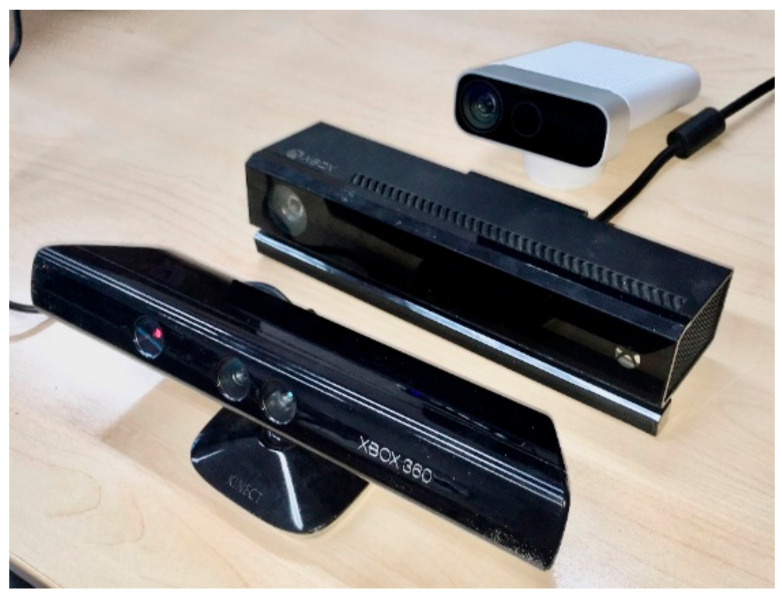
From left to right—Kinect v1, Kinect v2, Azure Kinect.

**Figure 2 sensors-21-00413-f002:**
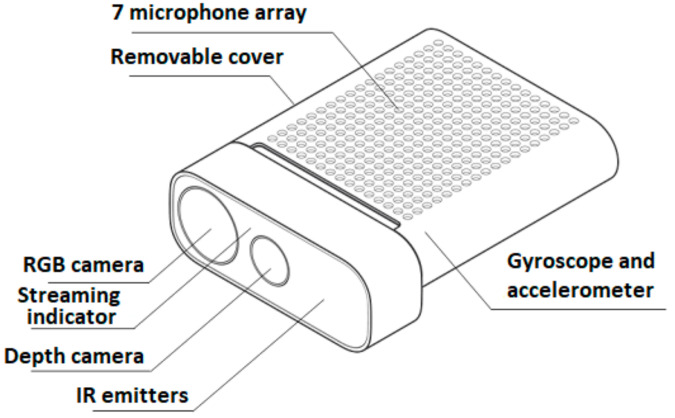
Schematic of the Azure Kinect.

**Figure 3 sensors-21-00413-f003:**
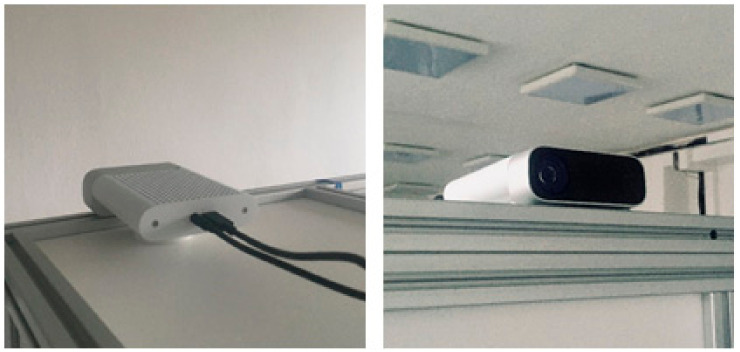
Sensor placement for testing purposes.

**Figure 4 sensors-21-00413-f004:**
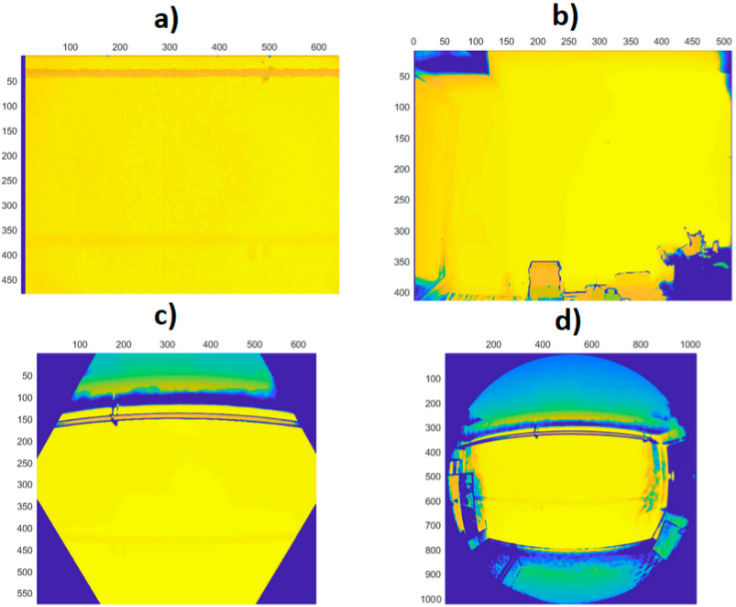
Typical data measurements acquired from Kinect v1 (**a**), Kinect v2 (**b**), Azure Kinect in narrow field-of-view (NFOV) binned mode (**c**), and Azure Kinect in wide field-of-view (WFOV) (**d**) sensors (axes represent image pixel positions).

**Figure 5 sensors-21-00413-f005:**
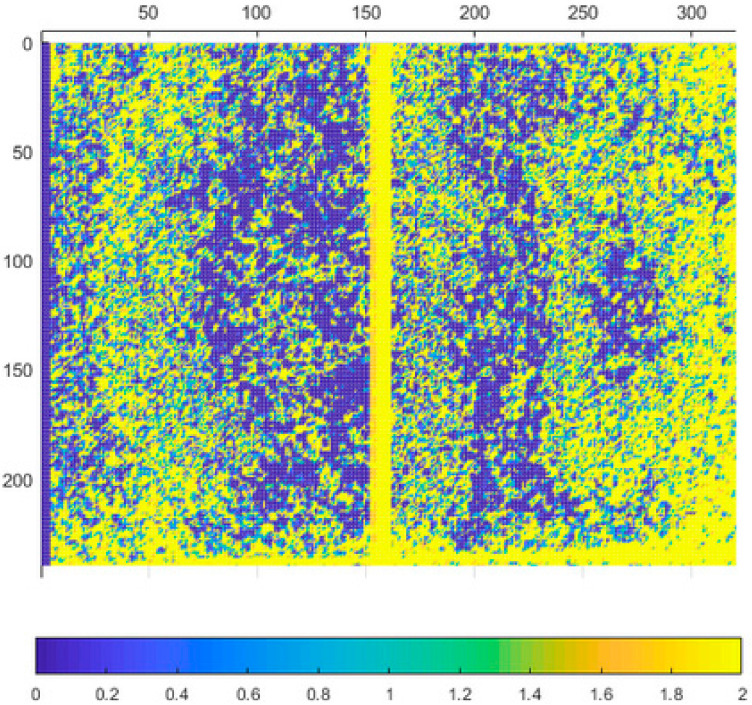
Typical depth noise of Kinect v1 in mm (values over 2 mm were limited to 2 mm for better visual clarity). Picture axes represent pixel positions.

**Figure 6 sensors-21-00413-f006:**
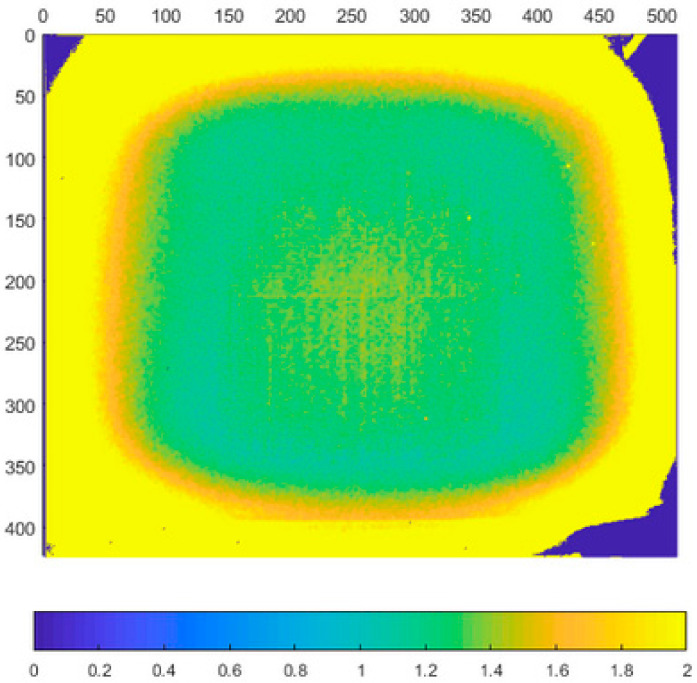
Typical depth noise of Kinect v2 in mm (values over 2 mm were limited to 2 mm for better visual clarity). Picture axes represent pixel positions.

**Figure 7 sensors-21-00413-f007:**
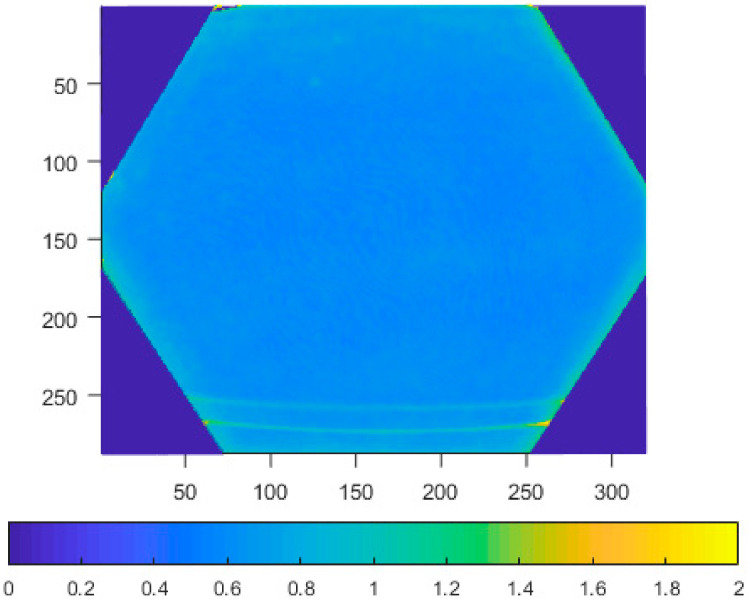
Typical depth noise of Azure Kinect in NFOV binned mode in mm (values over 2 mm were limited to 2 mm for better visual clarity). Picture axes represent pixel positions.

**Figure 8 sensors-21-00413-f008:**
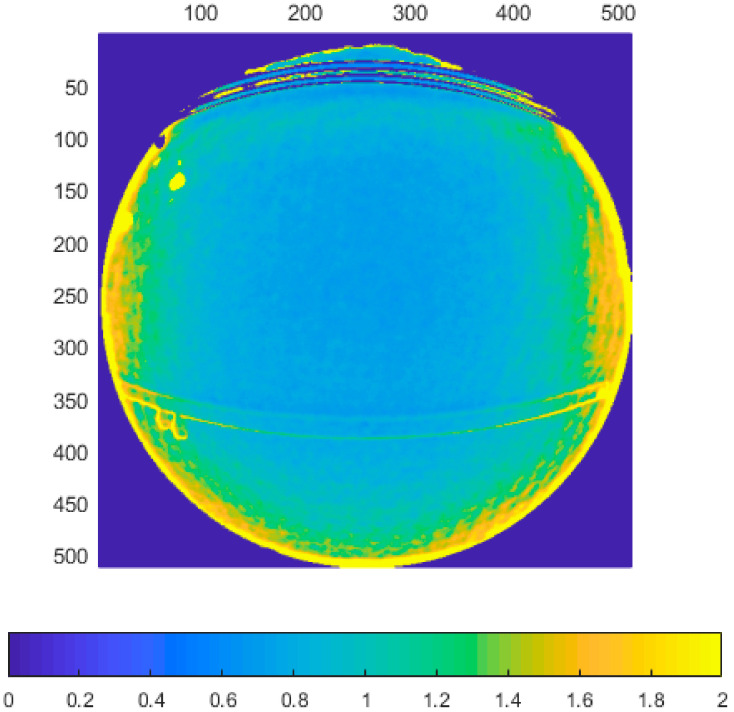
Typical depth noise of Azure Kinect in WFOV binned mode in mm (values over 2 mm were limited to 2 mm for better visual clarity). Picture axes represent pixel positions.

**Figure 9 sensors-21-00413-f009:**
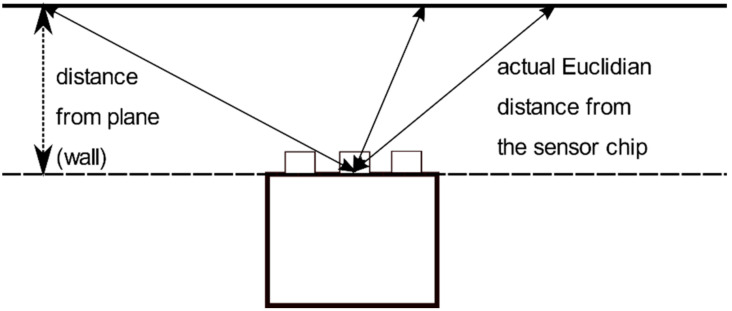
Plane vs. Euclidian distance of a 3D point form the sensor chip.

**Figure 10 sensors-21-00413-f010:**
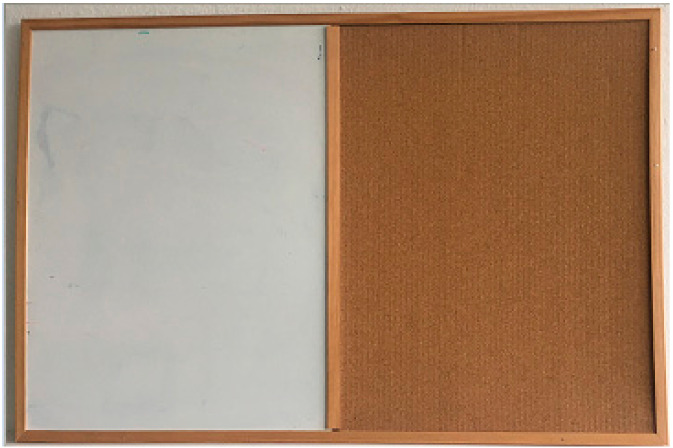
Test plate composed of plastic reflective material and cork.

**Figure 11 sensors-21-00413-f011:**
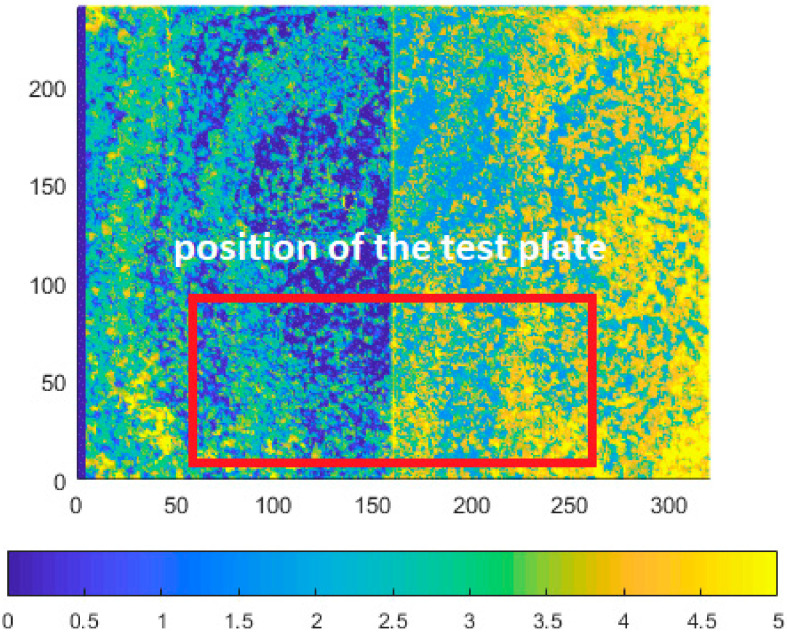
Depth noise of Kinect v1 with presence of an object with different reflectivity (values over 5 mm were limited to 5 mm for better visual clarity). Picture axes represent pixel positions.

**Figure 12 sensors-21-00413-f012:**
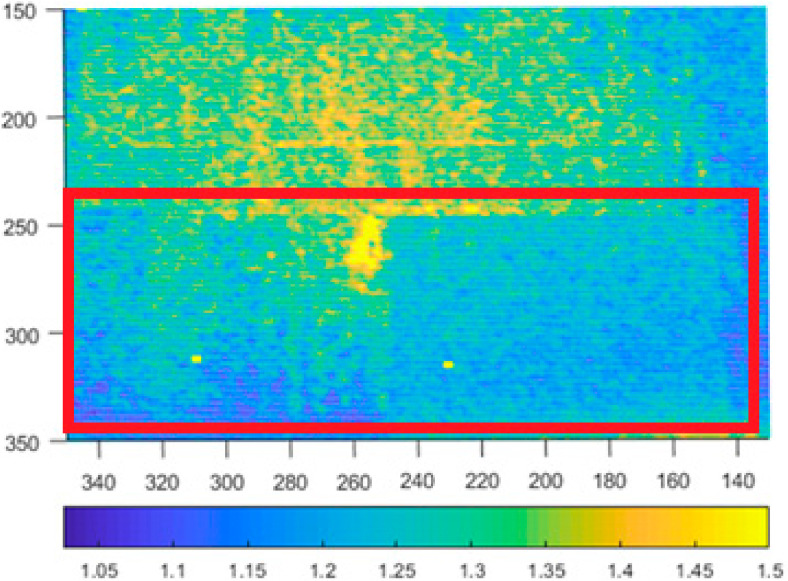
Depth noise of Kinect v2 with presence of an object with different reflectivity in the bottom area (values over 1.5 mm were limited to 1.5 mm for better visual clarity). Picture axes represent pixel positions.

**Figure 13 sensors-21-00413-f013:**
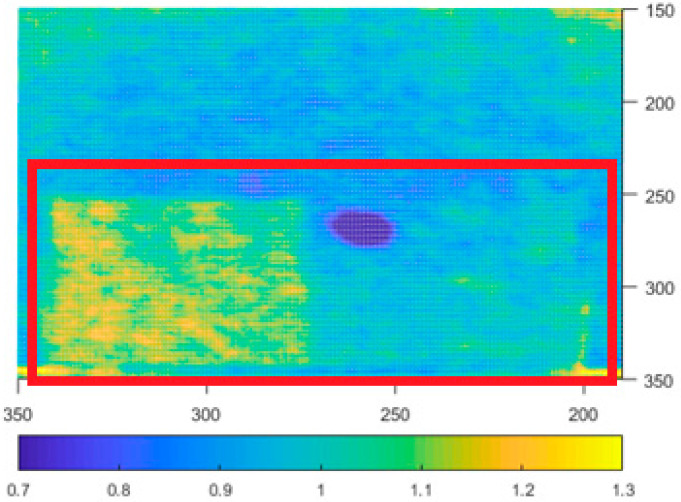
Depth noise of Kinect Azure with presence of an object with different reflectivity in the bottom area (values over 1.3 mm were limited to 1.3 mm for better visual clarity). Picture axes represent pixel positions.

**Figure 14 sensors-21-00413-f014:**
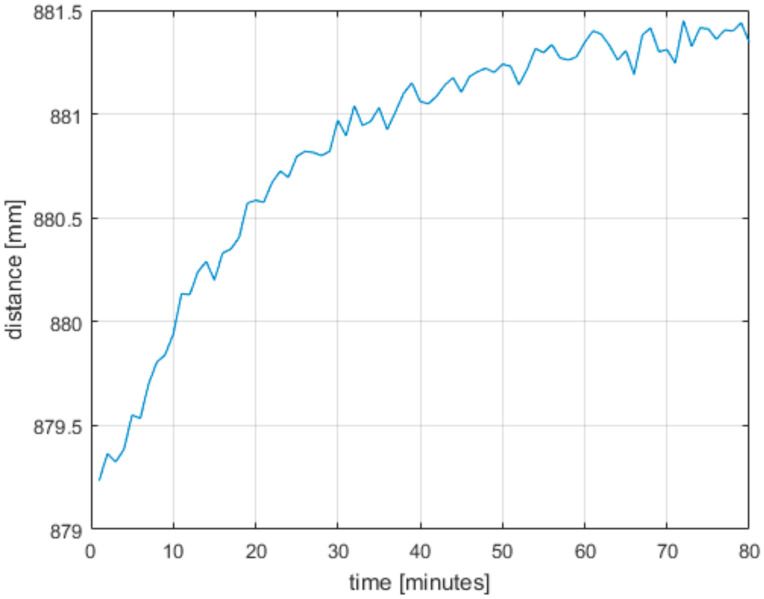
Measured distance while warming up the Azure Kinect. Each point represents the average distance for that particular minute.

**Figure 15 sensors-21-00413-f015:**
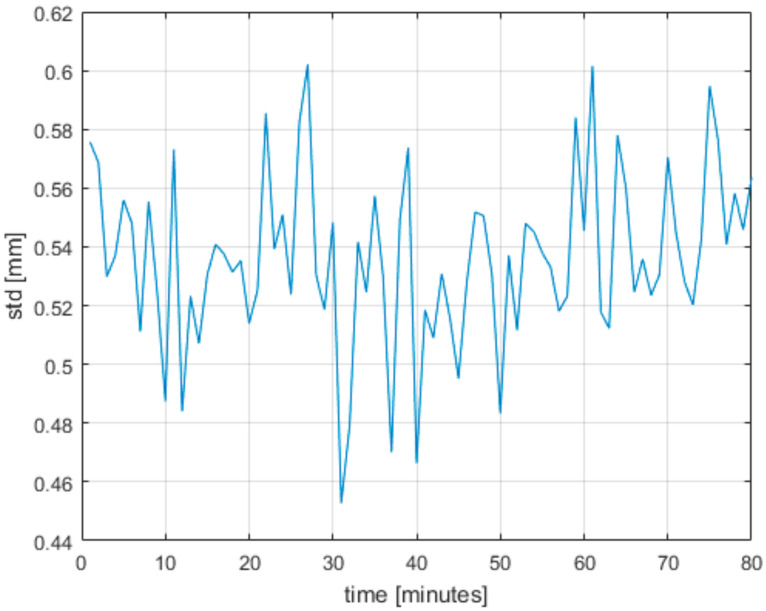
Measured standard deviation while warming up the Azure Kinect.

**Figure 16 sensors-21-00413-f016:**
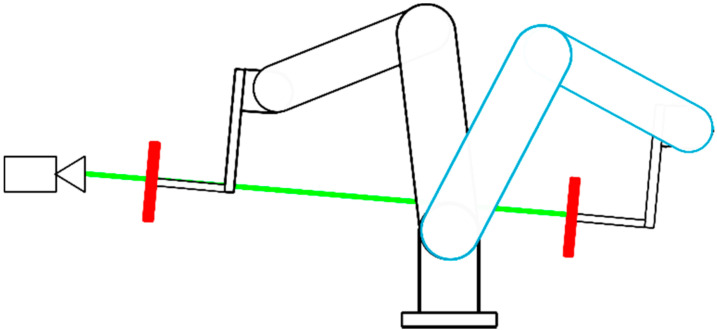
Scheme for accuracy measurements using robotic manipulator.

**Figure 17 sensors-21-00413-f017:**
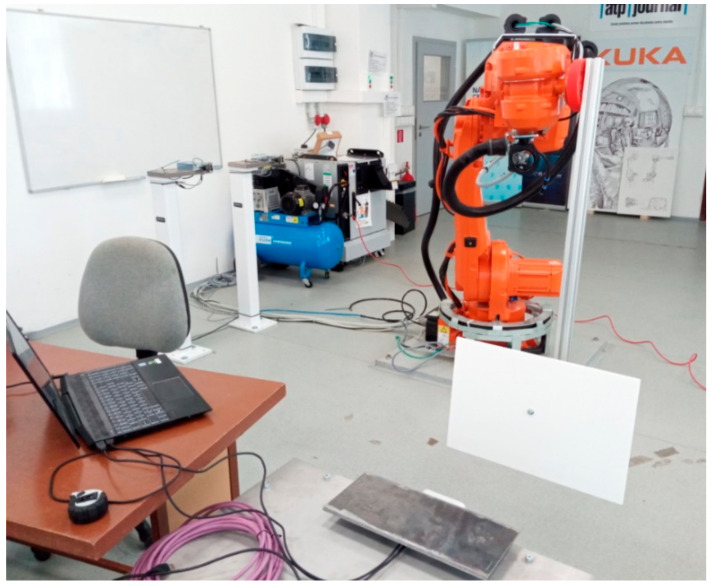
Picture of the actual laboratory experiment.

**Figure 18 sensors-21-00413-f018:**
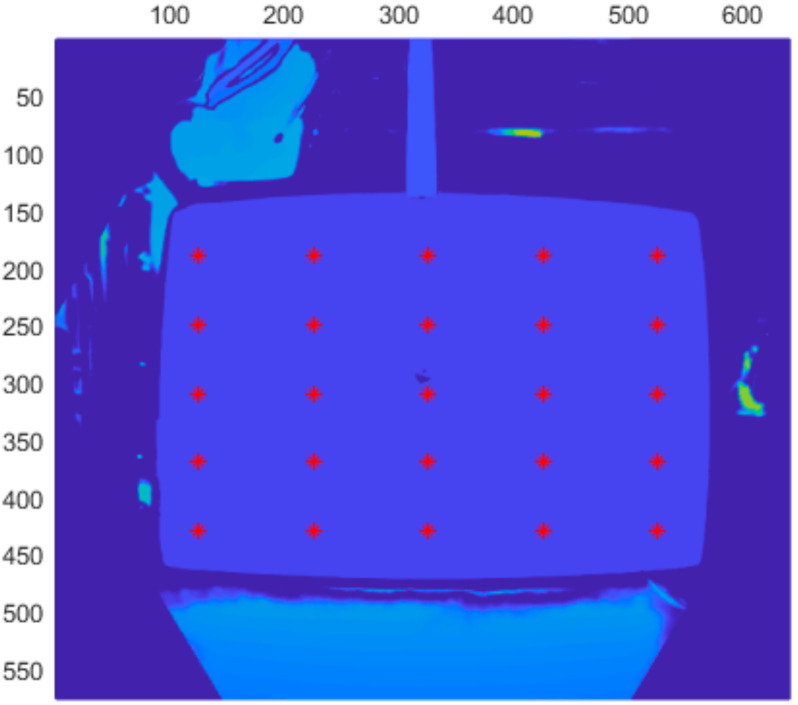
Selected depth points for fine tuning of the plate alignment. Picture axes represent pixel positions.

**Figure 19 sensors-21-00413-f019:**
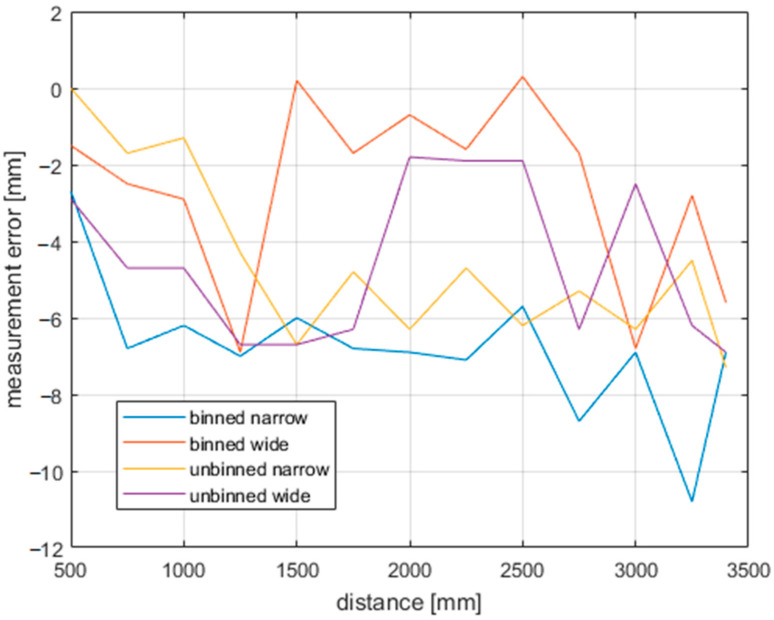
Accuracy of the Azure Kinect for all modes.

**Figure 20 sensors-21-00413-f020:**
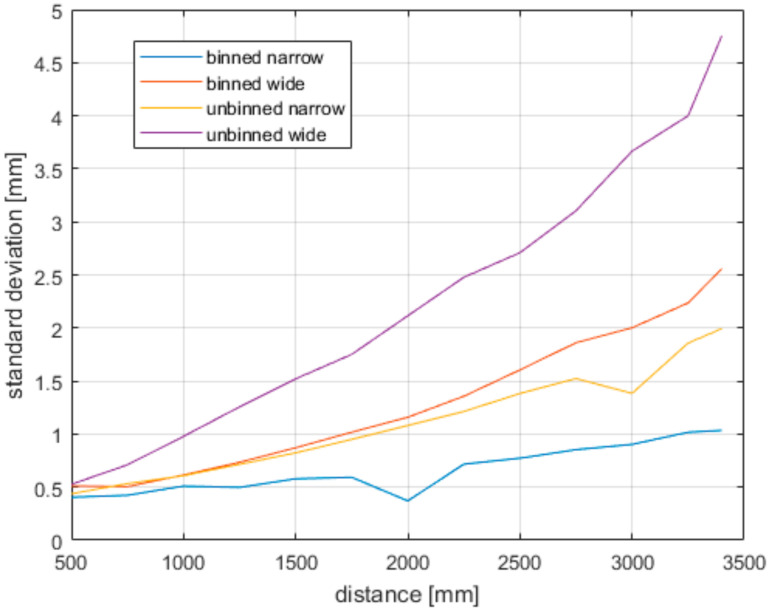
Precision of the Azure Kinect for all modes.

**Figure 21 sensors-21-00413-f021:**
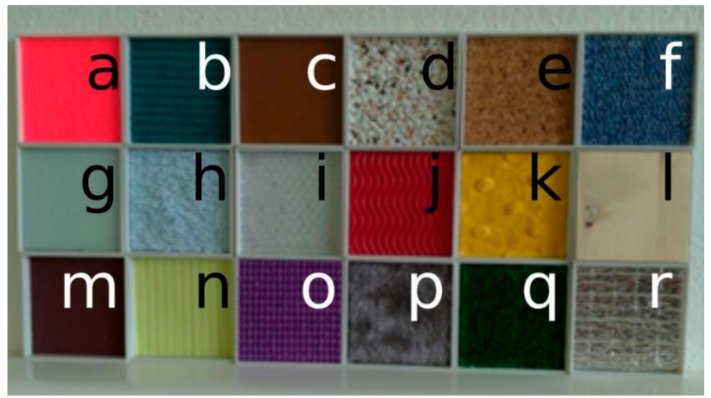
Layout of tested specimens: a—felt, b—office carpet (wave pattern), c—leatherette, d—bubble relief styrofoam, e—cork, f—office carpet, g—polyurethane foam, h—carpet with short fibres, i—anti-slippery matt, j—soft foam with wave pattern, k—felt with pattern, l—spruce wood, m—sandpaper, n—wallpaper, o—buble foam, p—plush, q—fake grass, r—aluminum thermofoil.

**Figure 22 sensors-21-00413-f022:**
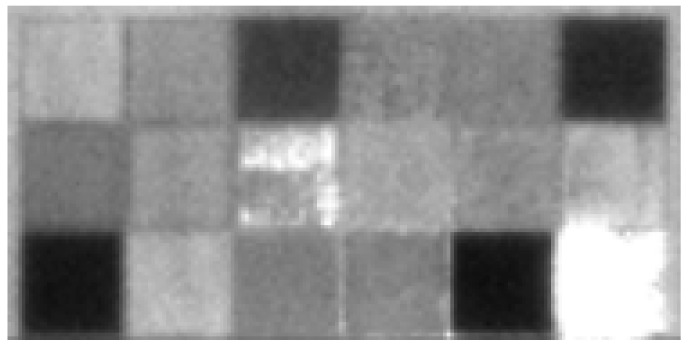
Infrared image of tested specimens.

**Figure 23 sensors-21-00413-f023:**
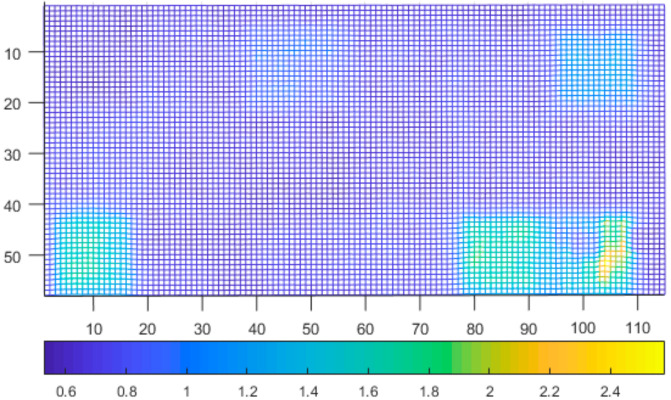
Standard deviation of tested specimens.

**Figure 24 sensors-21-00413-f024:**
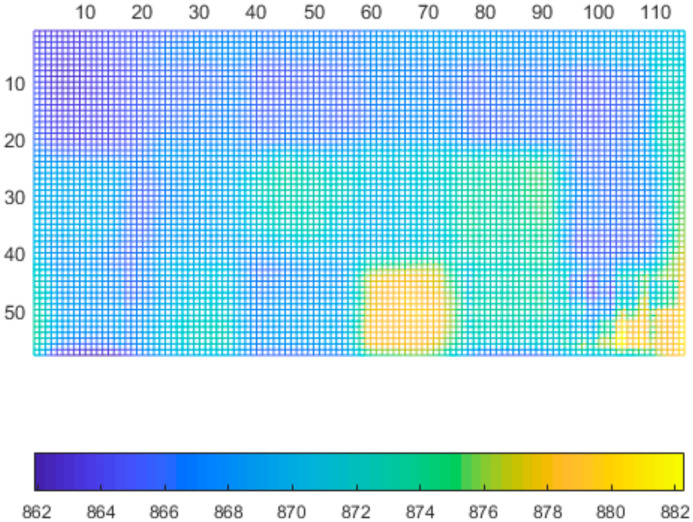
Average distance of tested specimens.

**Figure 25 sensors-21-00413-f025:**
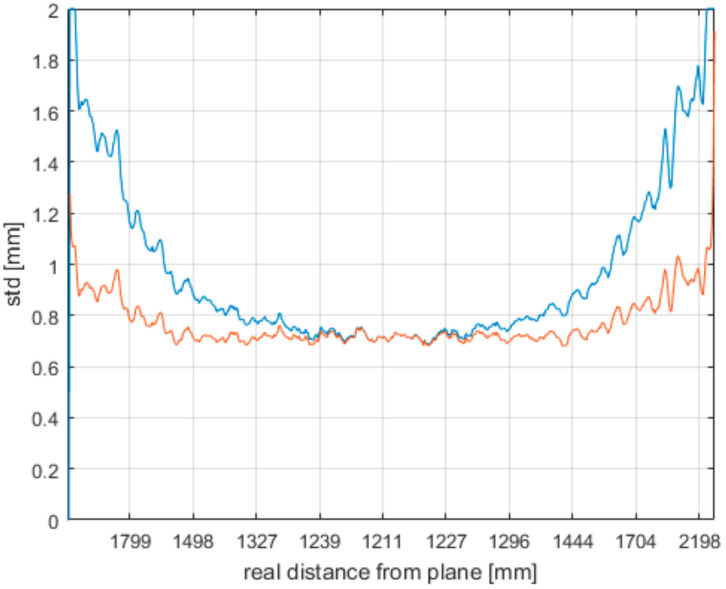
Correlation of noise and growing Euclidian distance from the sensor (blue curve—standard deviation of original data; orange curve—original data compensated with real Euclidian distance from the lens).

**Figure 26 sensors-21-00413-f026:**
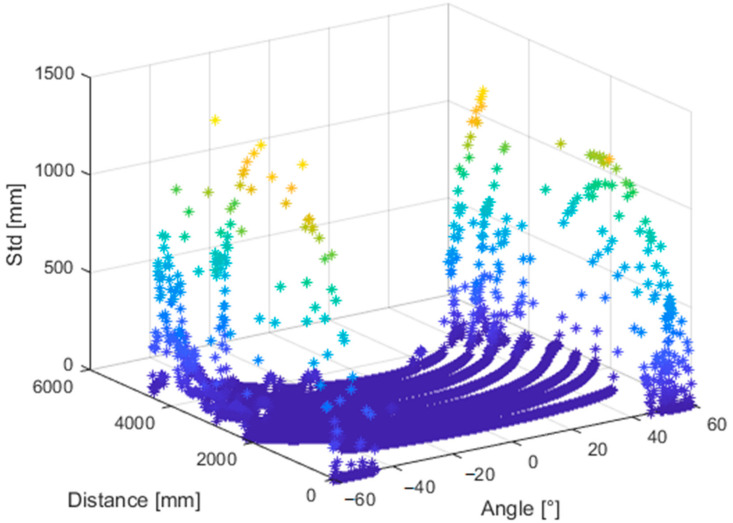
Standard deviation of noise of the Kinect Azure with respect to distance from the object (wall) and relative angle between the object and sensor.

**Figure 27 sensors-21-00413-f027:**
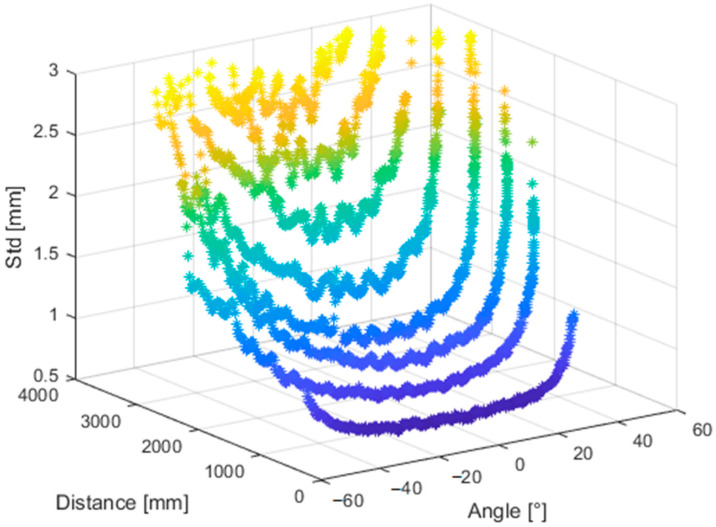
Truncated standard deviation of noise of the Kinect Azure with respect to distance from the object and relative angle of the object and sensor.

**Figure 28 sensors-21-00413-f028:**
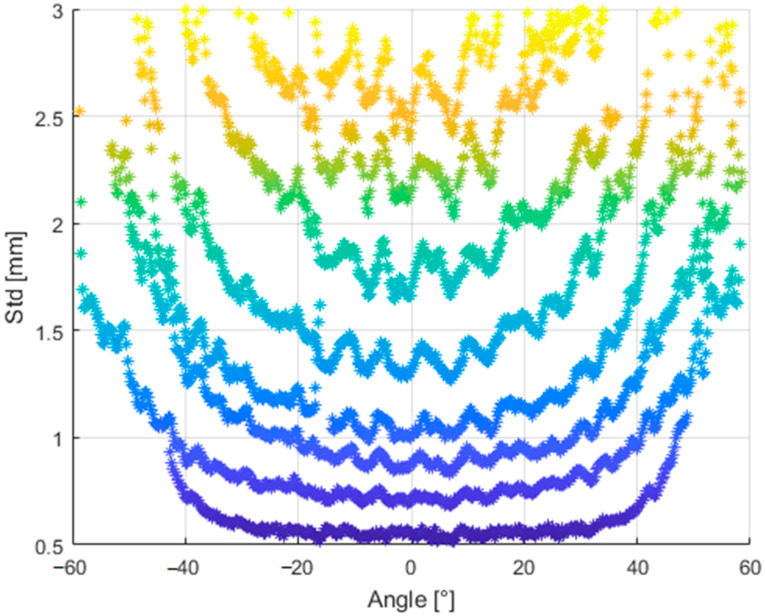
Standard deviation of noise of the Kinect Azure with respect to the relative angle of the object and sensor measured at different distances.

**Figure 29 sensors-21-00413-f029:**
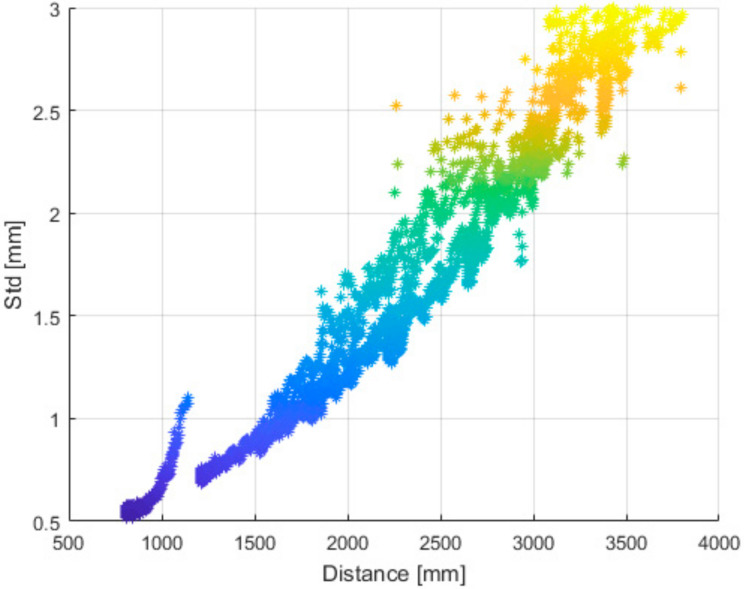
Standard deviation of noise of the Kinect Azure with respect to distance from the object with variable relative angles of the object and sensor.

**Figure 30 sensors-21-00413-f030:**
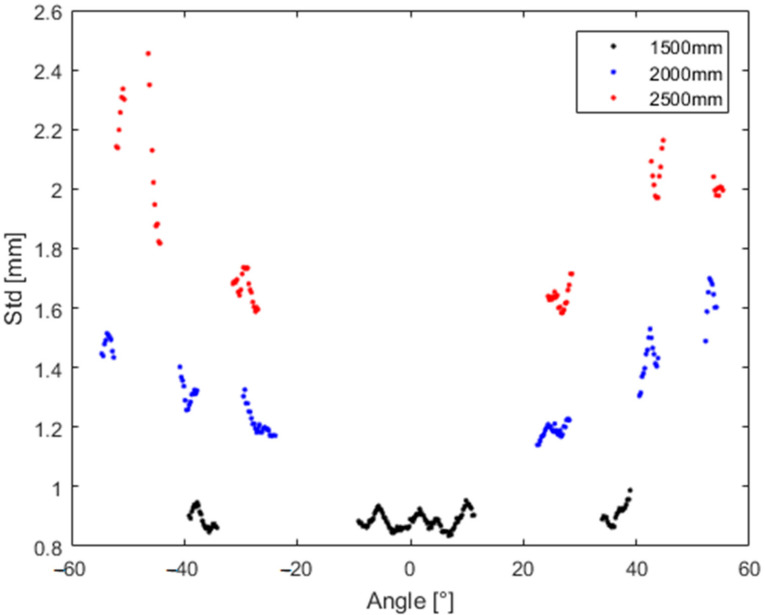
Standard deviation for particular identical distances with respect to the angle for which we measured.

**Figure 31 sensors-21-00413-f031:**
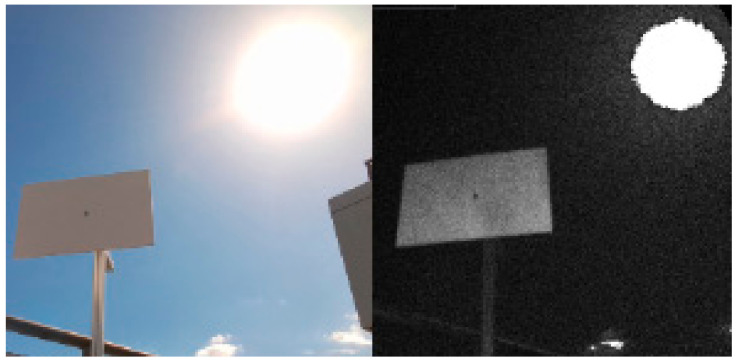
RGB and IR image of the first experiment scenario.

**Figure 32 sensors-21-00413-f032:**
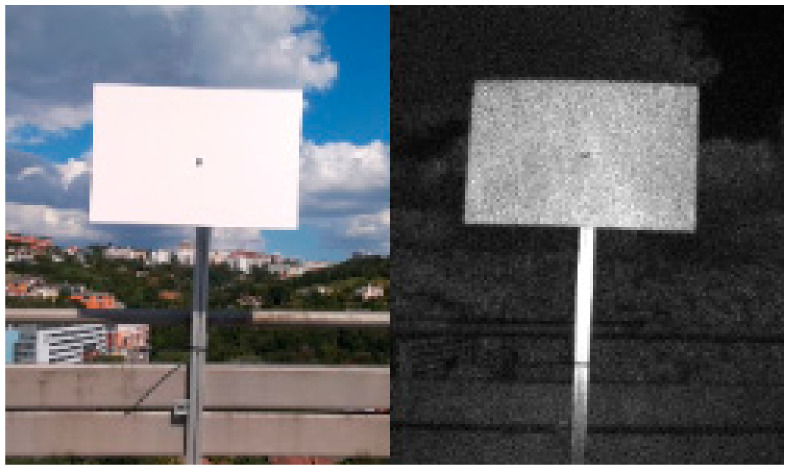
RGB and IR image of the second experiment scenario.

**Figure 33 sensors-21-00413-f033:**
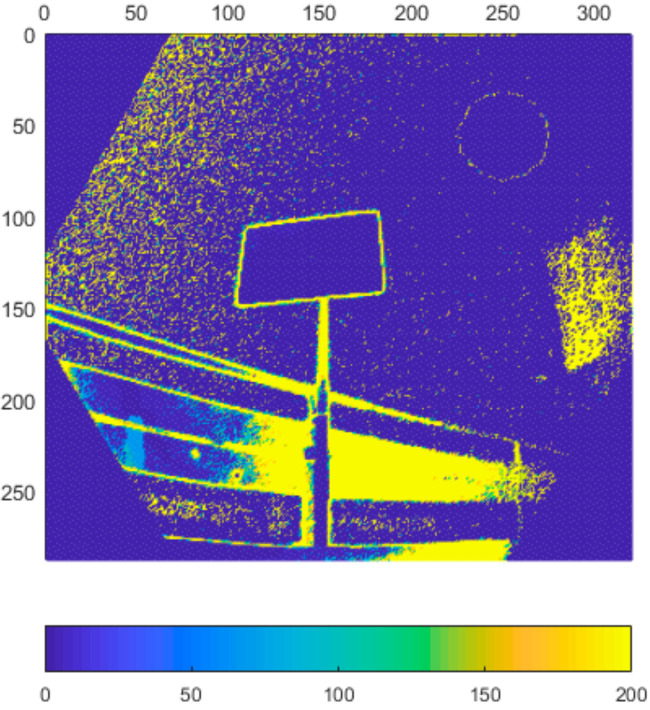
Standard deviation of binned NFOV mode limited to 200 mm (experiment 1).

**Figure 34 sensors-21-00413-f034:**
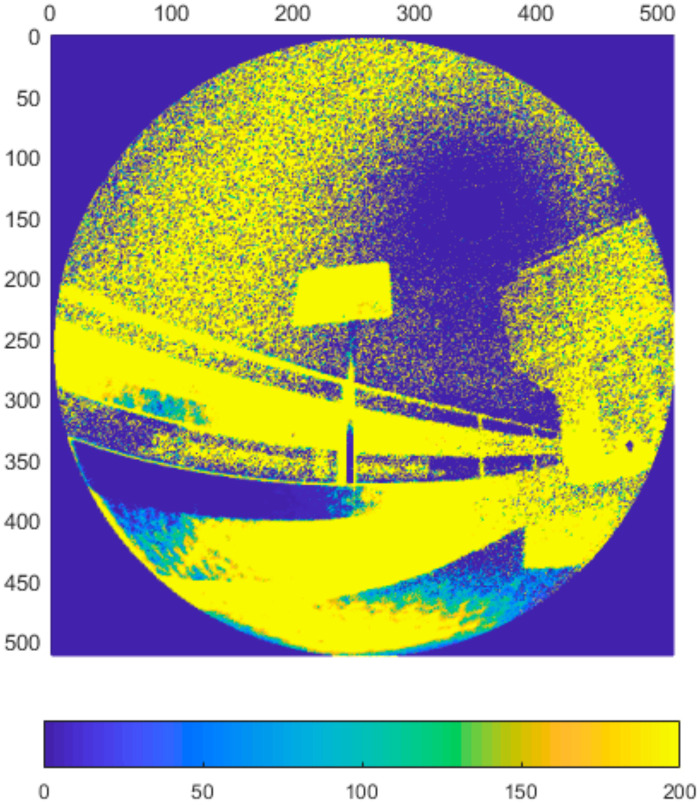
Standard deviation of binned WFOV mode limited to 200 mm (experiment 1).

**Figure 35 sensors-21-00413-f035:**
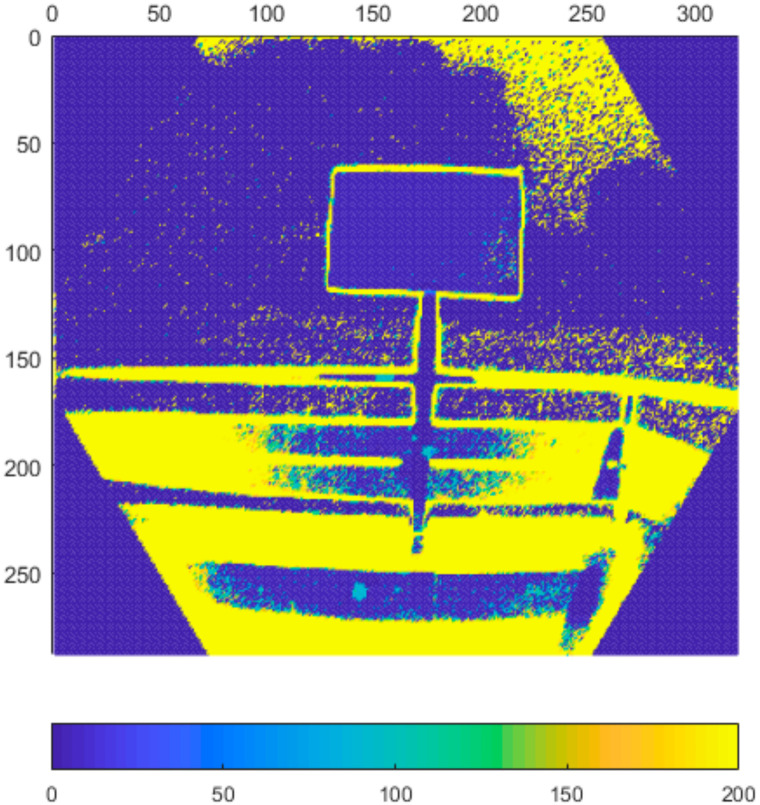
Standard deviation of binned NFOV mode limited to 200 mm (experiment 2).

**Figure 36 sensors-21-00413-f036:**
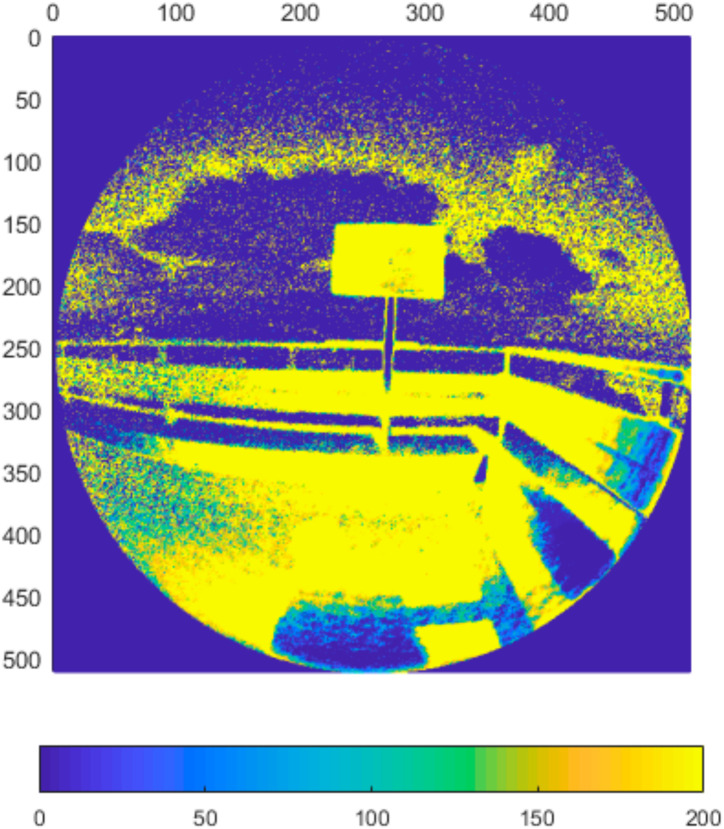
Standard deviation of binned WFOV mode limited to 200 mm (experiment 2).

**Figure 37 sensors-21-00413-f037:**
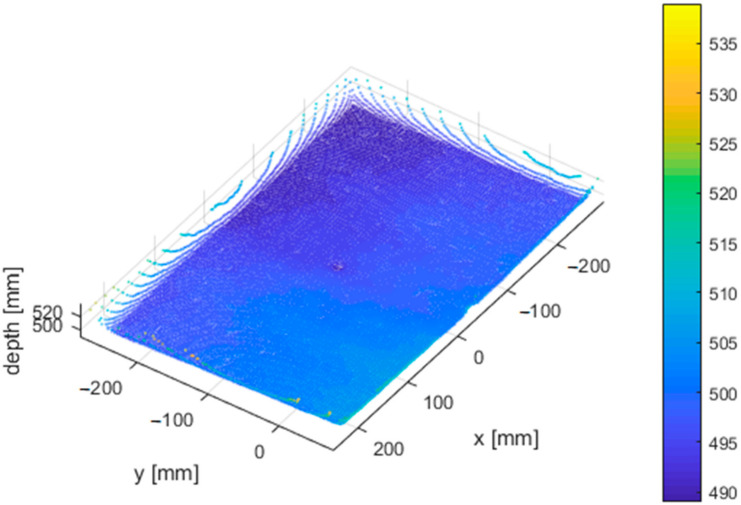
Demonstration of the flying pixel phenomenon–fluctuating depth data at the edge of the plate (front view, all values in mm).

**Figure 38 sensors-21-00413-f038:**
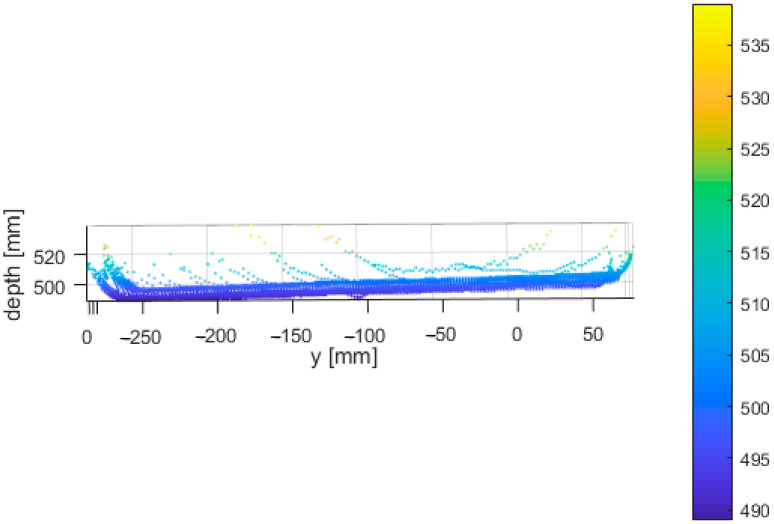
Demonstration of the flying pixel phenomenon–fluctuating depth data at the edge of the plate (side view, all values in mm).

**Table 1 sensors-21-00413-t001:** Comparison of the three Kinect versions.

	Kinect v1 [17]	Kinect v2 [26]	Azure Kinect
Color camera resolution	1280 × 720 px @ 12 fps 640 × 480 px @ 30 fps	1920 × 1080 px @ 30 fps	3840 × 2160 px @30 fps
Depth camera resolution	320 × 240 px @ 30 fps	512 × 424 px @ 30 fps	NFOV unbinned—640 × 576 @ 30 fps NFOV binned—320 × 288 @ 30 fps WFOV unbinned—1024 × 1024 @ 15 fps WFOV binned—512 × 512 @ 30 fps
Depth sensing technology	Structured light–pattern projection	ToF (Time-of-Flight)	ToF (Time-of-Flight)
Field of view (depth image)	57° H, 43° V alt. 58.5° H, 46.6°	70° H, 60° V alt. 70.6° H, 60°	NFOV unbinned—75° × 65° NFOV binned—75° × 65° WFOV unbinned—120° × 120° WFOV binned—120° × 120°
Specified measuring distance	0.4–4 m	0.5–4.5 m	NFOV unbinned—0.5–3.86 m NFOV binned—0.5–5.46 m WFOV unbinned—0.25–2.21 m WFOV binned—0.25–2.88 m
Weight	430 g (without cables and power supply); 750 g (with cables and power supply)	610 g (without cables and power supply); 1390 g (with cables and power supply)	440 g (without cables); 520 g (with cables, power supply is not necessary)

**Table 2 sensors-21-00413-t002:** Standard deviation of all sensors at different distances (mean value in mm).

	Kinect v1 (320 × 240 px)	Kinect v1 (640 × 480 px)	Kinect v2	Azure Kinect NFOV Binned	Azure Kinect NFOV Unbinned	Azure Kinect WFOV Binned	Azure Kinect WFOV Unbinned
800 mm	1.0907	1.6580	1.1426	0.5019	0.6132	0.5546	0.8465
1500 mm	3.1280	3.6496	1.4016	0.5800	0.8873	0.8731	1.5388
3000 mm	10.9928	13.6535	2.6918	0.9776	1.7824	2.1604	8.1433

**Table 3 sensors-21-00413-t003:** Standard deviation at different pixel locations on the chip (mean value in mm).

Parameter	54°	46°	39°	30°	16°	0°
Std (mm)	0.5957	0.6005	0.5967	0.6045	0.6172	0.6264

## Data Availability

All data generated or appeared in this study are available upon request by contact with the corresponding author.

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
