# Peer review of "Evaluation of the Azure Kinect and Its Comparison to Kinect V1 and Kinect V2"

_sensors, 2021, doi:10.3390/s21020413_

Round 1

Reviewer 1 Report

       The author provides a series of experiment results evaluating the performance of Azure Kinect in comparison with the previous generations of Kinect sensors. Exanimated aspects include warm-up time, precision, accuracy, color and material effect, precision variability, and outdoor performance.

       It’s good to see the author added material labels in figure 21, added the Euclidian distance experiment, and also added the flying pixel and multipath interference experiment.

       However, it must be stressed that the manuscript is lack of theoretical analysis although the author has conducted many experiments. The cause of performance improvement of Azure Kinect over Kinect v1 and v2 is little mentioned in the manuscript.

Author Response

Thank you for your time.

Reviewer 2 Report

The authors addressed all my concerns raised in my initial review. The modification of the manuscript is satisfactory. I do not have further comments or questions.

Author Response

Thank you for your time.

Reviewer 3 Report

The present review paper is devoted to the widely studied problems of depth sensors use in motion detection.  I have the following specific comments to the present submission:

Major comments:

  1. ABSTRACT: I suggest to enlarge the abstract with more details to methodology, results, and possible applications

  1. Section 1. INTRODUCTION: The topic of the paper is very important as it presents the new Azure Kinect which can replace the old Kinect v1 and v2. I suggest to include here selected relevant references related to applications in biomedicine including for instance the use of depth sensors in neurology and gait analysis:

[1] Prochazka A., Schaetz M., Tupa O., at al: The MS Kinect image and depth sensors use for gait features detection, Proceedings of the IEEE Int. Conf. on Image Processing, ICIP, Paris, France, 2014, pp. 2271-2274

[2] Stenert A., Sattler I., Otte K., at al: Using New Camera-Based Technologies for Gait Analysis in Older Adults in Comparison to the Established GAITRite System, Sensors, 20(1), 125, 2020

  1. Page 3, Table I: I suggest to mention the frame rate. Is it constant? How it is related the resolution? Was it tested?

  1. Section 3. COMPARISON OF ALL KINECT VERSIONS: The noise distribution should be statistically analyzed into more details.

  1. Section 4. EVALUATION OF THE AZURE KINECT: The experimental result related to the dependence of accuracy upon the warming is very important. The area of observation should be better specified? Is Fig. 14 related to the mean distance in selected area?

  1. Section 4.2 ACCURACY: The fact that the alignment of the dept and RGB sensor is not quite accurate is very important. How it is related to the distance?

  1. Section 5 CONCLUSION: I suggest to slightly enlarge this section and to mention possible applications

Minor comments:

  1. I suggest to organize better images to save the space, some figures can be on the same line side by side (Figs. 7 and 8; or Figs. 11, 12 and 13; or Figs. 14, 15 for instance)

  1. References should be revised to be in the same format and small mistakes corrected

Round 2

Reviewer 1 Report

There is minor changes comparing to the previous manuscript. It must be stressed that the manuscript is lack of theoretical analysis although the author has conducted many experiments. The cause of performance improvement of Azure Kinect over Kinect v1 and v2 is little mentioned in the manuscript.

Author Response

The Azure Kinect is a proprietary sensor, and Microsoft does not provide detailed information. Therefore, it would be only a speculation to determine the cause of the performance improvement; therefore, a deeper theoretical analysis is
not possible.

This manuscript is a resubmission of an earlier submission. The following is a list of the peer review reports and author responses from that submission.

Round 1

Reviewer 1 Report

This manuscript reports the comparison of Azure Kinect and previous models of the Microsoft Kinect line of sensors, Kinect v1 and v2. The authors describe some general features of all three sensors as well as the output, which includes sensor depth data, noise to distance correlation, and noice to reflectivity correspondence. Finally, the authors evaluate the performance of the Azure Kinect with a focus on warm-up time, accuracy, precision, reflectivity, precision variability, and sensitivity in outdoor environments. With all the experiments, the authors conclude that the new Azure Kinect is better than both previous versions and provide suggestions of the application scenario of Azure Kinect based on their listed advantages and disadvantages.

Although this work contains many experimental results and information, the manuscript leaves a number of flaws in the motivation and discussion. I recommend acceptance after major revision. Below are some detailed comments.

  1. In the introduction, the authors mention "there have been hundreds of paper written and published" to discuss the technical features of Kinect v1 and v2. They focus on the analysis and evaluation of the new Azure Kinect as it is released recently in 2019. This statement itself is not clear as it fails to address the significance of the study. An open question is why would a potential user refer to this paper for understanding major technical features of Azure Kinect given there is a manual that can be downloaded anytime for free?
  2. Many terms or objects studied in this work lacks clear definition and quantification. For example, how to define "warm-up time", why is a measured distance 2 mm above the starting value considered "warmed up". The subtitles of "accuracy" and "precision" are misleading, the authors may consider swap them with "positional accuracy" and "repeatability of positional accuracy measurements", respectively. Or consider merge 4.3 to 4.2 which makes better sense. "Reflectivity of materials" requires quantification before the conclusion of "less reflective materials have higher standard deviation" can be drawn. Also, some discussion on the tested specimens can be added in a separate table. After reading the entire section 4.4, the only material that the authors even mention is "aluminum foil".
  3. Almost all of the figures require more detailed captions, scale bars for illustration of dimensions, and names of the axes. Without them, the figures provides minimum useful information to the potential readership.
  4. Many abbreviations are not well defined when they first appear. For example, SLAM, HMI, HCI, etc.
  5. The existing tables have no table legends. Information in table 1 is missing.
  6. Many inconsistencies in the context. For example, Fig. 15 "standard deviation" is abbreviated as std.; line 361, it is shortened to std. dev.

Reviewer 2 Report

       The author provides a series of experiment results evaluating the performance of Azure Kinect in comparison with the previous generations of Kinect sensors. Exanimated aspects include warm-up time, precision, accuracy, color and material effect, precision variability, and outdoor performance.

       Although the performance evaluation of Azure Kinect is practically valuable, the manuscript is lack of theoretical analysis. Neither the influence mechanisms between the factors (e.g. color and material) and the performance indicators (precision and accuracy), nor the cause of performance improvement of Azure Kinect over Kinect v1 and v2 is discussed thoroughly in the manuscript.

  1. In section 4.2, the accuracy examination result has no comparison with the previous Kinect sensors.
  2. In section 4.4, the author should separate the two variables, color(reflectivity) and material, to give a firm conclusion on the effect of color(reflectivity) and material towards precision and accuracy separately. Also, for the ease of comparison, material labels in figure 21-24 is expected.
  3. Line 260 says “we devised an experiment specifically for each of these factors”, but experiment measuring the effect of the first aspect (line 252-253) cannot be found.
  4. Reference[2] made comparison of Kinect v1 and v2 in the aspects of flying pixel and multipath interference. The author may consider including these experiments as well for the completeness of comparison.